# Decision-Making in the Pediatric Emergency Department—A Survey of Guidance Strategies among Residents

**DOI:** 10.3390/children9081197

**Published:** 2022-08-09

**Authors:** Sebastian Gaus, Jeremy Schmidt, Paul Lüse, Winfried Barthlen, Eckard Hamelmann, Hendrik Vossschulte

**Affiliations:** 1Pediatric Emergency Department, University Children Hospital Bielefeld (EvKB), 33617 Bielefeld, Germany; 2Department of Pediatrics, University Children Hospital Bielefeld (EvKB), 33617 Bielefeld, Germany; 3Department of Pediatric Surgery, University Children Hospital Bielefeld (EvKB), 33617 Bielefeld, Germany

**Keywords:** pediatric emergency department, decision-making, guidance strategies, smartphone application, internet platform, digital age

## Abstract

(1) Introduction: Working in an emergency department requires fast and straightforward decisions. Therefore, decision guidance represents an essential tool for successful patient-centered care. Beyond the residents’ own knowledge and experience, printed books have been the primary source of information in the past. The aim of this study was to discover which strategies current residents use the most and to identify alternative quick reference strategies in the digital age. (2) Materials and Methods: This study analyzed the responses of a short questionnaire directed at 41 residents in a single pediatric emergency department (32 pediatric and 9 pediatric surgery residents) over a period of one month. (3) Results: Thirty-three (80.5%) residents answered the entire questionnaire. Strikingly, responses indicated that printed books are still pivotal in guiding decision-making. In addition, the acquisition of information via computers or smartphones plays an increasing role. However, the opinion and council of the attending physician is still of great value to the residents and is not to be underestimated. Overall, most of the residents would prefer to have access to a specially designed smartphone application. (4) Conclusions: Certainty and validity are essential in decision-making in a pediatric emergency department. Although printed books and attending physicians are still considered as reliable sources of information, internet-based information plays an increasing role. In order to provide the best up-to-date and most recent information, a validated and consistently updated smartphone application could be a useful option.

## 1. Introduction

In a (pediatric) emergency department, the quick accessibility of qualified information is a pivotal factor for the successful patient-centered care [1]. Higher awareness and preparation for critical pediatric situations result in lower mortality [2]. Time-sensitive and critical situations include anaphylaxis [3], burns [4], sepsis [5], shock [6], and choosing the right medication for various events [7,8]. Therefore, pediatric and pediatric surgery residents require fast and easy access to databases and may use the internet or specialized smartphone applications more often than established printed books. The digital transformation has impacted many occupational fields over the last decades. In medicine, the use of digital applications has been accelerated inherently due to the COVID-19 pandemic. At the same time, medical knowledge is becoming progressively more comprehensive, while the internet provides faster and easier availability and accessibility for the public [9]. Medical platforms and smartphone applications for providing this information have been developed, in part specifically for health professionals [10], showing positive impacts on clinician behavior and patient effects [11]. However, it is difficult to keep track of these rapid developments, especially in terms of the quality assurance and reliability of new platforms/app operators or certifications. This poses questions regarding whether the eHealth literacy of residents is high enough to ensure that the quality of the retrieved information of any given internet source meets a high-quality standard leading to subjective certainty in decision guidance [12]. To guarantee the best standard care, it is necessary to provide not only fast but also reliable information about potentially time-sensitive and critical health conditions or problems. This study was designed as a pilot in-house investigation in an interdisciplinary pediatric emergency department of a pediatric university hospital. The underlying question was how and where pediatric and pediatric surgery residents gather the necessary medical information within their daily working routine. The primary aim was to evaluate whether ubiquitously available and accessible smartphones and their applications would emerge as major instruments for decision guidance.

## 2. Materials and Methods

Forty-one residents of different educational levels working in an interdisciplinary pediatric emergency department (32 in the pediatric and 9 in the pediatric surgery department) of a university hospital were asked to answer an online survey consisting of four questions. The questionnaire was self-developed by the authors and set up using SurveyMonkey©, an online data analysis platform. Answers had to be given via multiple choice and free text. The primary request as well as two further reminders were sent via e-mail. The data acquisition time spanned 4 weeks. The questionnaire focused on the application of different digital and non-digital sources of information during working hours. Distinctions between the two departments considering gender, age, or education were not made. The questionnaire was anonymous and did not allow the possibility to draw any personnel-related conclusions. 

The questionnaire contained the following four questions:How certain are you regarding your therapeutic decisions before you retrieve information or ask somebody else for help?What kind of quick reference do you preferentially use as a decision guidance in uncertain situations during your work in the emergency department?The options to answer the second question were the following:AMBOSS^©^: a Germany-based online learning platform for medical students and residents (www.amboss.com, accessed on 30 July 2022), fee requiredUpToDate^©^: an English-based online learning platform of the Wolters Kluwer Health Division (www.uptodate.com, accessed on 30 July 2022), provided to hospital employeesGoogle: an English-based online search engine (www.google.com, accessed on 30 July 2022), free of chargeGoogle Scholar: an English-based online platform for scientific literature research (www.scholar.google.de, accessed on 30 July 2022), free of chargee-books: private/offered by the hospital employerprinted booksThe residents were asked to rate the following statements:“I use the online journals offered for free by the employee hospital online library”“I use specific smartphone applications”“I ask colleagues of a similar level of education”“I ask the senior/attending physician”“I ask the nursing staff”What kind of smartphone applications do you generally use?Would you use a comprehensive smartphone application more often if offered?

## 3. Results

Thirty-three (80.5%) of a total of forty-one residents answered the questionnaire. Out of the resulting 33 sets of answers, 24 were derived from the pediatric and 9 from the pediatric surgery department. 

**First question**: How certain are you regarding your therapeutic decisions before you retrieve information or ask somebody else for help?

All 33 residents answered the question. Figure 1 visualizes the results. 

This shows that most of the residents (69.7%) felt “certain” or “absolutely certain” in their therapeutic decisions, whereas less than one third (30.3%) were “not so certain”.

**Second question**: What kind of quick reference do you preferentially use as a decision guidance in uncertain situations during your work in the emergency department?

The second question was answered by 32 residents; one resident (3.0%) skipped the question. Multiple answers were allowed. Table 1and Figure 2 show the results.

In order to further stratify the answers, the two options “always” and “often” were categorized as “positive” and the two options “seldom” and “never” were categorized as “negative”. Figure 2 shows the results.

Positive answers were most commonly given for the attending physician (72.8%), printed books (66.6%), colleagues (63.6%), and Google (57.6%), whereas choices like UpToDate^©^ (42.5%), asking the nursing staff (42.4%), AMBOSS^©^ (39.5%), specific smartphone applications (33.3%), free online journals (21.2%), and e-books (18.2%) had the fewest positive results.

**Third question**: What kind of smartphone applications do you generally use?

The third question was answered in free text by 30 respondents (90.9%) and skipped by 3 (9.1%) residents. The author suggested no specific applications. The named smartphone applications were put in categories considering their medical content: drug reference, calculator (for drugs, percentiles etc.), AMBOSS^©^, applications designed by other hospitals, and others (see Table 2). Multiple answers were possible.

The most frequently used applications were those for drug reference (39.4%). Medical calculators were the second most commonly used applications (36.4%). The AMBOSS^©^ application was mentioned by 24.2% of the residents. Other named applications were medical press sources, Pschyrembel^©^ (a German-based medical databank, also available as a smartphone application), the German Standing Committee on Vaccination (STIKO), the AO Surgery Reference of the AO Foundation (an international non-profit network of surgeons), and UpToDate^©^. The applications of other hospitals mentioned were those from a pediatric hospital in Switzerland.

**Fourth question**: Would you use a comprehensive smartphone application more often if offered?

The fourth question was positively (summarized for “always” and “preferentially”) answered by 27 (81.8%) of the residents and as “preferentially no” by only 3 (9.1%). No resident answered “never” and 3 (9.1%) skipped this question (see Figure 3).

## 4. Discussion

Nearly 70% of the residents in the pediatric emergency department displayed a high degree of certainty regarding their medical decisions. Although the residents would appreciate a comprehensive smartphone application as a useful tool in their daily routine, the most utilized sources in their daily practice remained printed books and colleagues with advanced experience.

Decision-making processes have been evaluated in several medical fields. While many practitioners in oncology, for example, use mobile devices, most of them attest that the latter only have a small impact on their daily practice [13,14]. The same study also stresses the necessity of the reliability of these applications [13,15]. 

In an evaluation among more than 300 physicians in the United Kingdom in 2012, nearly 80% used applications (e.g., formularies, calculators, clinical decision tools) to gain improved access to information, and more than 50% used these applications to improve their decision-making. It was also discovered that only 23% of the study participants compiled a risk assessment of the applications prior to using them [16]. This low percentage might raise some concern in the light of some of these applications suggesting a certain treatment strategy to satisfy commercial interests [17].

To our best knowledge, the subjective “certainty” of decision-making is not something that has been studied or discussed within medical literature up to this point. Few studies pay attention to the efficacy of medical staff working in an emergency department. Dowd et al. (2005) defined efficacy in a pediatric emergency department as a quantitative parameter (“patients per hour”) and found dependencies on experience (more patients per hour were associated with more training years) and higher specialization (with residents of emergency medicine seeing more patients per hour than pediatricians, and pediatricians seeing more patients per hour than family medicine residents) [18]. Trotzky (2021) defined efficacy as the length of stay in an adult emergency department and found that patients seen by emergency medicine specialists had the shortest length of stay. Furthermore, the decision time of these specialists was shorter compared to that of residents or of internal medicine or general surgery specialists [19]. They concluded that expertise was the crucial factor in an emergency department. This underlines the theory that a higher certainty in decision processes can be achieved with increasing expertise, and that it can be improved by undergoing special training. However, Zundel (2017) also noted the lack of educational training for pediatric surgery residents [20].

Popularity and quick availability seem to play the biggest role in choosing an internet-based reference tool. An indirect problem while choosing a reference tool could lie in the quality of information provided. Search engines might not be focused on prioritizing well-ordered, certified information for medical staff. Also, the use of learning platforms depends on their accessibility and availability in the native language. In our case, AMBOSS^©^ was not provided free of charge by the employer and could only be used by residents willing to pay for the platform or application access. UpToDate^©^ was provided by the hospital for free, but was not excessively utilized in the investigated emergency room, as demonstrated by the results.

As a matter of fact, our survey’s emergency department provided a selection of printed textbooks on pediatrics and pediatric surgery for its staff. The convenience of having information at hand might be one reason for this tool’s high approval rate. Another factor could be the widespread conception that printed books equal valid knowledge. E-books, however, were not provided by the hospital. That could explain why the residents did not draw on this tool as much. The low utilization rate of online journals could be explained by a lower degree of assumed usefulness in emergency situations. Although these media contain condensed information, residents might feel that they are difficult to screen for reliable content when information is required on short notice.

Printed books (and e-books) face the general concern of possibly outdated or even inaccurate information by not being able to keep up with the rapid development of medical information. A survey among German radiologists showed that 67% of 104 participants used e-books as reference tools. However, among these residents, 87.5% also criticized e-books as simply being electronic copies of common books. Another 57% of the residents preferred e-books instead of medical smartphone applications [21]. Maybe these concerns best explain why, in our survey, most residents expressed the highest degree of trust in their attending physicians’ advice. Another reason could be, that specifically considering the subject pediatrics, given it’s high case-by-case variability and many influencing factors in decision-making (e.g., age group differences, third party anamnesis, parents’ desires), the personalized advice from an attending physician provides more certainty than rather general information provided by books or platforms. However, when considering orphan diseases combined with an anticipated further knowledge increase and possibly enhanced platforms and apps, in some situations a shift to external decision supports, which can collectively display large international data sets, seems reasonable.

In 2012, a survey among young physicians in the United Kingdom showed that applications with a high acceptance rate among residents are often designed for drug calculation and drug references [22]. Our results confirmed these findings. Specific drug administration applications can reduce medication errors and the time needed for drug preparation, and they are especially relevant in a pediatric setting with its inherent individual drug indication and dosage [23]. 

Our survey’s residents used a variety of different smartphone applications. Problematically, most of these medical applications were designed for adult patients and for use in low-income countries lacking adequate medical support [15]. Therefore, they have rarely been evaluated among pediatricians or pediatric surgeons [23,24]. Once released, an application of medical guidelines (in this case for pediatric emergencies) may have more than 2500 users in more than 100 countries after a period of three months [25]—a reference to the enormous dissemination potential of medical guidelines via smartphone applications. From the authors’ point of view, an application that fulfils almost all of the residents’ requirements and, in addition, allows for quick access to standard operating procedures would be highly appreciated. In the above-mentioned survey among radiologists, 41% of the residents described a distinct work simplification by using medical applications, whereas 31% felt a modest improvement, 16% saw no improvement, and 12% communicated an absolute improvement [21].

Obtaining the best medical information within the shortest amount of time in acute situations within an emergency department is a challenge. As most people obtain their information via the world wide web, the use of smartphone applications seems to be a logical consequence. However, which one is the best? In 2014, more than 300 smartphone applications specifically developed for emergency medicine were available [26]. However, only a low percentage of applications seem to be truly relevant for medical professionals [27]. In a German interdisciplinary statement on the usage of cognitive tools within pediatric emergency care, the authors state that, for instance, smartphone applications for emergency drugs and their calculation are mostly not certified by state agencies or reviewed by scientific societies. In the past years, research has led to the development of quality criteria and official reimbursement for health apps in Germany; however, specific recommendations for the pediatric age group are missing [28,29]. 

Furthermore, the usage of medical apps depends on a functional network access [30]. This may be a problem in some hospitals due to security concerns or structural conditions. It could also explain why, in our study, smartphone applications were only utilized regularly by about one-third of the participating residents. Further reasons may be a lack of knowledge and insight into existing applications due to insufficient awareness, teaching, and training. 

## 5. Conclusions

While attending physicians seem to be the most trusted source of information regarding quick reference or guidance in emergency situations, there is also a need to trust in oneself and be self-confident in making basic decisions. Classical printed books seem to be a major source for the residents’ quest for information. However, to stay in line with the current state of knowledge, easily accessible and regularly updated web or smartphone applications could be the next steps in the ongoing training of medical residents. These applications should incorporate subject-specific guidelines and in-house standard operating procedures. We believe that this process should be monitored by the scientific community and subject-specific societies, which should work on standardized pediatric-specific criteria or at least prepare robust recommendations.

## 6. Limitations

This study included only a small sample due to the nature of it being a single center survey within one pediatric emergency department. Thus, center-specific local guidelines, established processes, and availability of resources may have influenced the results. A gender assignment could not be achieved because the employee representation did not approve of such an approach. The use of in-house guidelines in terms of standard operating procedures as an answering option was not evaluated due to it not being part of this survey. Furthermore, the questionnaire was self-developed and not validated beforehand, which may have resulted in measurement inaccuracies. 

## Figures and Tables

**Figure 1 children-09-01197-f001:**
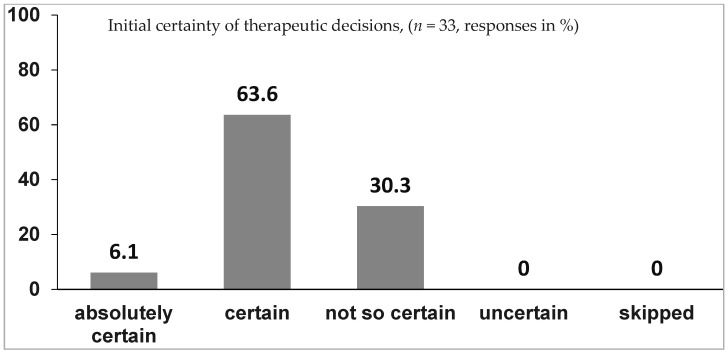
Responses to question 1.

**Figure 2 children-09-01197-f002:**
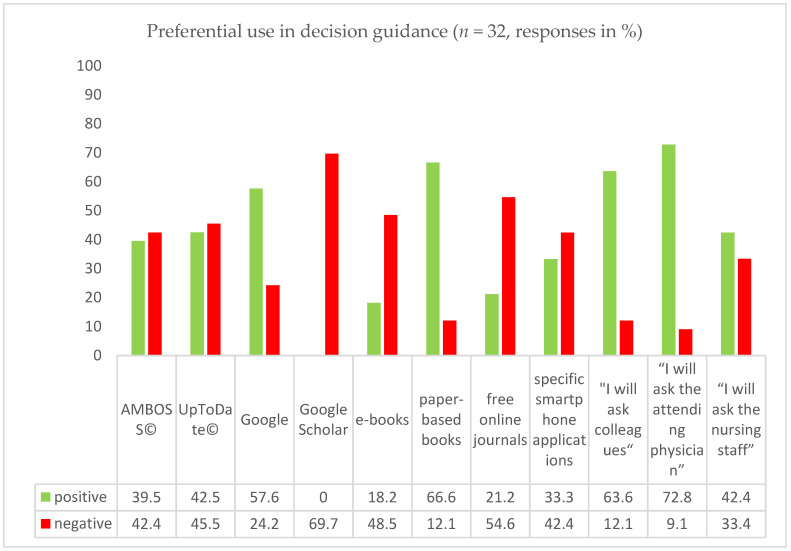
Positive and negative responses to question 2.

**Figure 3 children-09-01197-f003:**
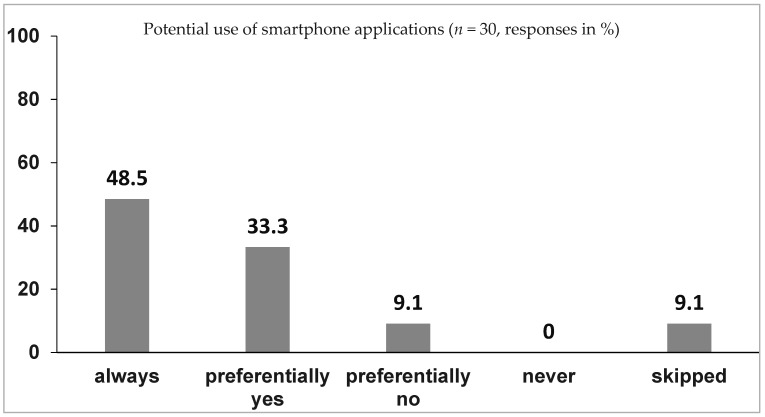
Responses to question 4.

**Table 1 children-09-01197-t001:** What kind of quick reference do you preferentially use as a decision guidance in uncertain situations during your work in the emergency department? Missing data were defined as sets of answers where one answering option was skipped (1).

Answering Options (% of *n* = 33)	Total Answers	Always	Often	Seldom	Never	Missing Data
AMBOSS^©^	28 (84.8)	2 (6.1)	12 (33.4)	6 (18.2)	8 (24.2)	5 (15.2)
UpToDate^©^	30 (90.9)	3 (9.1)	12 (33.4)	13 (39.4)	2 (6.1)	3 (9.1)
Google	27 (81.8)	4 (12.1)	15 (45.5)	7 (21.2)	1 (3.0)	6 (18.2)
Google Scholar	23 (69.7)	0 (0.0)	0 (0.0)	7 (21.2)	16 (48.5)	10 (30.3)
e-Books (private/hospital)	22 (66.7)	3 (9.1)	3 (9.1)	9 (27.3)	7 (21.2)	11 (33.3)
printed books	26 (78.8)	4 (12.1)	18 (54.5)	4 (12.1)	0 (0.0)	7 (21.2)
free online journals	25 (75.8)	1 (3.0)	6 (18.2)	5 (15.2)	13 (39.4)	8 (24.2)
specific smartphone applications	25 (75.8)	3 (9.1)	8 (24.2)	8 (24.2)	6 (18.2)	8 (24.2)
“I will ask colleagues”	25 (75.8)	3 (9.1)	18 (54.5)	4 (12.1)	0 (0.0)	8 (24.2)
“I will ask the attending physician”	27 (81.8)	5 (15.2)	19 (57.6)	3 (9.1)	0 (0.0)	6 (18.2)
“I will ask the nursing staff”	24 (72.7)	3 (9.1)	10 (33.3)	9 (27.3)	2 (6.1)	9 (27.3)

**Table 2 children-09-01197-t002:** Responses to question 3.

Application Category	*n* (%)
drug reference	13 (39.4)
calculator	12 (36.4)
AMBOSS^©^	8 (24.2)
others	10 (30.3)
applications of other hospitals	4 (12.1)

## Data Availability

The dataset analyzed for the study will be available from the corresponding author on reasonable request.

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
