# Peer review of "Decision-Making in the Pediatric Emergency Department—A Survey of Guidance Strategies among Residents"

_children, 2022, doi:10.3390/children9081197_

Round 1

Reviewer 1 Report

Thank you for the opportunity to review this interesting article on a relevant and under-researched field. I have some comments.

#I would suggest changing the title to: "Decision making in the pediatric emergency department: a survey of guidance strategies among residents"

#Please remove "of maximum care" after university hospital as this is redundand information.

#One major issue of the study is that the use of in-house guidelines and SOP´s was not included in the survey which may cause a relevant bias. This is, however, addressed by the authors in the limitations section.

#The paper is well presented but there are some language and style issues throughout the manuscript that need to be fixed in a revision.

Author Response

Dear reviewer,   thank you very much for this helpful review, the positive feedback, and constructive comments. Please find attached our responses to your comments.

Kind regards,
Sebastian Gaus and Jeremy Schmidt 

Reviewer 2 Report

Dear Authors,

This is an interesting manuscript. I have some suggestions and comments.

Abstract - Line 20 - Please check the percentage - 33 (88.5%).

Introduction: Line 44-48: There is a contradiction here regarding the quality of informations provided by the internet. A reference is needed, also. 

Materials and Methods 

-Who created the questionnaire? It was used before? It is validated?

-I think you should choose between Figure 1 and Table 1. Presenting both is redundant. Same for Table 2 and Figure 2, Table 3 and Figure 4, Table 4 and Figure 5.

Discussions: I believe that no application can replace the experience of a primary care physician, especially in the case of children, with the particularities related to age and each special condition. I woul like to read more about this kind of direct learning in the Discussion chapter. 

Best regards. 

Author Response

Dear reviewer,

thank you very much for this helpful review and the constructive comments. Please find attached our responses to your comments.

Kind regards,

Sebastian Gaus and Jeremy Schmidt

Round 2

Reviewer 1 Report

I would like to thank Dr Gaus and colleagues for the revision of the manuscript. All comments have been addressed appropriately and - together with the important points raised by the reviewer colleague - the paper now presents considerably improved. In my humble opinion, this work may be another piece of the puzzle in this underresearched field and certainly will add to the literature. Best wishes

Reviewer 2 Report

Dear Authors,

Thank you for the answers to all my concerns. I consider that the manuscript can now be published. 

Best regards.